# Relationships between total reserve and financial indicators of Bangladesh: Application of generalized additive model

**Md. Sifat Ar Salan[1], Mahabuba Naznin[2], Bristy Pandit[2], Imran Hossain Sumon[1], Md. Moyazzem Hossain** **[1,3]\*, Mohammad Alamgir Kabir[1], Ajit Kumar Majumder[1]**

**1** Department of Statistics, Jahangirnagar University, Savar, Dhaka, Bangladesh, **2** Department of Statistics, Mawlana Bhashani Science and Technology University, Santosh, Tangail, Bangladesh, **3** School of Mathematics, Statistics, and Physics, Newcastle University, Newcastle upon Tyne, United Kingdom

\* hossainmm@juniv.edu

## Abstract

### Background

The reserve of a country is a reflection of the strength of fulfilling its financial liabilities. However, during the past several years, a regular variation of the total reserve has been observed on a global scale. The reserve of Bangladesh is also influenced by several economic and financial indicators such as total debt, net foreign assets, net domestic credit, inflation GDP deflator, net exports (% of GDP), and imports of goods and services (% of GDP), as well as foreign direct investment, GNI growth, official exchange rate, personal remittances, and so on. Therefore, the authors aimed to identify the nature of the relationship and influence of economic indicators on the total reserve of Bangladesh using a suitable statistical model.

### Methods and materials

To meet the objective of this study, the secondary data set was extracted from the World Bank's website which is openly accessible over the period 1976 to 2020. Moreover, the model used the appropriate splines to describe the non-linearity. The performance of the model was evaluated by the Akaike information criterion (AIC), Bayesian information criterion (BIC), and adjusted R-square.

### Results

The total reserve of Bangladesh gradually increased since 2001, and it reached its peak in 2020 which was 43172 billion US dollars. The data were first utilized to build a multiple linear regression model as a base model, but it was later found that the model has severe multicollinearity problems, with a maximum value of VIF for GNI of 499.63. Findings revealed that total debt, inflation, import, and export are showing a non-linear relationship with the total reserve in Bangladesh. Therefore, the authors applied the Generalized Additive Model (GAM) model to take advantage of the nonlinear relationship between the reserve and the selected covariates. The overall response, which is linearly tied to the net foreign asset in

**Data Availability Statement:** This study is based on the secondary dataset. The data is freely available on World Development Indicators of

World Bank database. Data can be access through the link: https://data.worldbank.org/country/BD.

**Funding:** The author(s) received no specific funding for this work.

**Competing interests:** The authors have declared that no competing interests exist.

the GAM model, will change by 14.43 USD for every unit change in the net foreign asset. It is observed that the GAM model performs better than the multiple linear regression.

## Conclusion

A non-linear relationship is observed between the total reserve and different economic indicators of Bangladesh. The authors believed that this study will be beneficial to the government, monetary authorities also to the people of the country to better understand the economy.

## Introduction

The total reserve plays a vital role in the economy of a country and the global financial market receives indications about that nation's creditworthiness and reliability of monetary policy from it. Total reserve refers to all of a bank's deposits with a Federal Reserve bank as well as the cash, coins, and gold it holds in its safe to satisfy all of its liabilities [1]. The reserve of Bangladesh is not noticeably bigger than what is needed, according to several reserve adequacy metrics based on international best practices [2]. Only when one takes into account the current characteristics of the reserves benchmark, which may be appropriate for the nation as well as its financial system, do the reserves of the country stand higher than the necessary level. A satisfactory level of total reserves is required for the stabilization of the development process. It is a challenge for developing countries to increase total reserves in order to make timely payments for imports [3]. The formulation and implementation of present and future macroeconomic policies aimed at achieving a trade balance heavily rely on foreign reserves [4]. In other words, these proponents hold the view that a robust level of reserves will make the country appear financially responsible and creditworthy in the eyes of other countries, creditors, and donors [5]. Either from a political or economic perspective, a very sensitive indicator for any government is to maintain a satisfactory level of international reserve in its coffer.

Total reserve is the combination of foreign exchange reserve and domestic reserve. The term foreign exchange reserve refers to the supply of foreign currency currently being kept by the central bank of a country [6, 7]. Foreign exchange reserves were defined by International Monetary Fund (IMF) in 2000 in this way "foreign reserves are defined as the external stock of asset, which is available to the country's monetary authorities to cover external payment imbalances or to influence the exchange rate of the domestic currencies through intervention in exchange market or for other purposes" [8]. For Bangladesh's economy, an unprecedented rise in remittances in recent years has resulted in reserve accumulation [9]. According to a Heritage Foundation report, Bangladesh's economic freedom score is 56.4, making it's economy the 122nd freest in the 2020 Index, and it has increased by 0.8 points overall, led by a higher score for property rights. Bangladesh is ranked 29th among 42 countries in the Asia-Pacific region, and its overall score is well below the regional and world averages [10].

A previous study highlighted that Bangladesh's long-term reserve demand policies are heavily influenced by the current account's vulnerability and exchange rate flexibility [11]. Researchers ascertain the factors that influence Bangladesh's demand for international reserves [12]. The reserve of a country reflects the overall strength of that country to combat its sudden financial hazard of it. However, no study has been found that considers the most recent data as well as considering the comparison of the linear and non-linear model in the context of Bangladesh. To fill up this gap, it is necessary to identify the key determinants of the total reserve

of Bangladesh. Therefore, the authors aimed to identify the nature of the relationship and influence of economic and financial indicators including total debt, net foreign assets, net domestic credit, inflation GDP deflator, net exports (% of GDP), and imports of goods and services (% of GDP), as well as foreign direct investment, GNI growth, official exchange rate, personal remittances, and so on the total reserve. As the primary goal of our study is to find the factors affecting the total reserve of Bangladesh, it will help the government let along the monetary authority of the country focus on how to improve the reserve. A suitable model for the economic data beyond the econometric discipline is also a matter of investigation in this study. The authors believed that this study will be beneficial to the government, monetary authorities also to the people of the country to better understand the economy.

## Literature review

Nowadays, predicting the reserve of a country is a challenging task based on financial indicators. Over the last couple of decades, the rapid increase in gross financial flows, the resulting outsize external stocks in relation to GDP, and the growth of domestic financial systems have made the international reserve more vulnerable [13]. However, reserves can be largely attributed to current patterns of expanding trade openness and increased sensitivity to financial shocks by emerging nations [14]. But if the factors affecting the total reserve are unknown then it will be difficult to understand the changes over time and forecasting. Over the past decade, developing nations' holdings of international reserves, particularly India, have grown quickly [15]. According to World Bank Report (2014), the world's total reserves have more than doubled in the shortest amount of time here total reserves also include foreign exchange reserves. This sudden rise in forex reserves motivates researchers to investigate the determinants and variables that are effectively affecting reserves [16]. There is an association between foreign exchange reserves, the current account balance to GDP ratio, the debt to GDP ratio, the domestic private sector's share of GDP, the exchange rate, inflation, per capita GDP, and the real interest rate [17, 18]. Economic growth is linked to the energy consumption of a country [19] and the $CO_2$ emissions from diverse sources are linked to both short- and long-term economic growth [20, 21], financial development adversely impacted the environment as well [22–25]. Moreover, the natural resources of a country also influenced its financial development [26]. Economic development particularly in Bangladesh is hugely relied on the advancement of agricultural growth [27]. Researchers pointed out that different crops production in Bangladesh has risen significantly after 2010 [28–35], however, the distribution of natural resources and climate change impacted the sustainable agricultural production [36–39] that will be a cause of impact on the total reserve in future.

Researchers conducted a study based on the nineteen member nations of the Asia Cooperation Dialogue to determine the dynamics between the financial factors, and they found that the total reserves, financial development, consumption of renewable energy, trade openness, tourism, and improved sanitation all closely interact with one another [3]. A similar study was done by Disyatat and Mathieson (2001) for fifteen countries in Asia and Latin America and submitted that the volatility of the exchange rate is an important determinant of reserves accumulation and that the financial crisis of the late 1990s produced no structural breaks [40]. According to a study conducted in Indonesia, the variable foreign loans has a positive and large impact on foreign exchange reserves, meaning that if foreign loans rise, reserves will rise dramatically. Foreign loans that describe financial conditions in terms of external financing. In addition, the import variable significantly and negatively affects foreign exchange reserves [41]. In the Indian context, the opportunity cost has a much greater influence on reserve demand than does reserve volatility, which is a sharp contrast to the overall picture in

emerging market economies [42]. Moreover, variables including imports, broad money, exchange rate volatility, and interest rate difference had an impact on India's reserve [15, 43]. According to a prior study, factors like short-term external debt and GDP have an impact on foreign exchange reserves over the long term, as well as inflation. The short-term impact of the exchange rate, however, is favorable for India's foreign exchange reserves [44].

## Methods and materials

### Data and variables

The authors used annual time series data for Bangladesh from 1976 to 2020 for the eleven variables, including total reserve as the response and total debt, net foreign assets, net domestic credit, inflation GDP deflator, net exports (% of GDP), and imports of goods and services (% of GDP), as well as foreign direct investment, GNI growth, official exchange rate, and personal remittances as covariates. The secondary data is obtained from the World Development Indicators of World Bank database [45] and data is publicly available and one can be access through the link: https://data.worldbank.org/country/BD. The selection of carraraites is based on the existing literature review, data availability, and self-efficacy of the authors. The government and those who make decisions can focus on the factors that we have identified as crucial for Bangladesh's reserve. Additionally, the government and central bank should put in place a number of policies to reduce rising import costs, such as restricting the imports of non-essential items, while the dollar reserves are being drained. Fig 1 presents the study's research framework. To identify the research gap, a thorough assessment of the literature related to the study of total reserves is undertaken as a first step. The authors then used a time series plot to observe the trend of the total reserves in Bangladesh. Step 1 additionally verifies the factors' identification and the direction of their link to the total reserve. In phase two, the appropriate model is determined. The final step involves performing the diagnostic assessment and making predictions [Fig 1].

### Non-Linear regression

In statistics, nonlinear regression is a type of regression analysis in which observational data are represented by a function that depends on one or more independent variables. In order to fit the data, a method of successive approximations is used. A mathematical model known as nonlinear regression uses a produced line to solve an equation for specific data. As is the case with a linear regression that uses a straight-line equation, nonlinear regression shows association using a curve, making it nonlinear in the parameter. A simple nonlinear regression model is expressed as follows:

$$Y = f(X, \beta) + \varepsilon$$

where $X$ is a vector predictor, $\beta$ is a vector of parameters, $f(.)$ is the known function, and $\varepsilon$ is the error term.

### Polynomial regression

In statistics, polynomial regression is a form of regression analysis in which the relationship between the independent variable $X$ and the dependent variable $Y$ is modeled as a nth degree polynomial in $X$. In many settings, a linear relationship may not hold. In this case, we might propose a quadratic model of the form,

$$y = \beta_0 + \beta_1 x + \beta_2 x^2 + \ldots + \beta_p x^p + \varepsilon$$

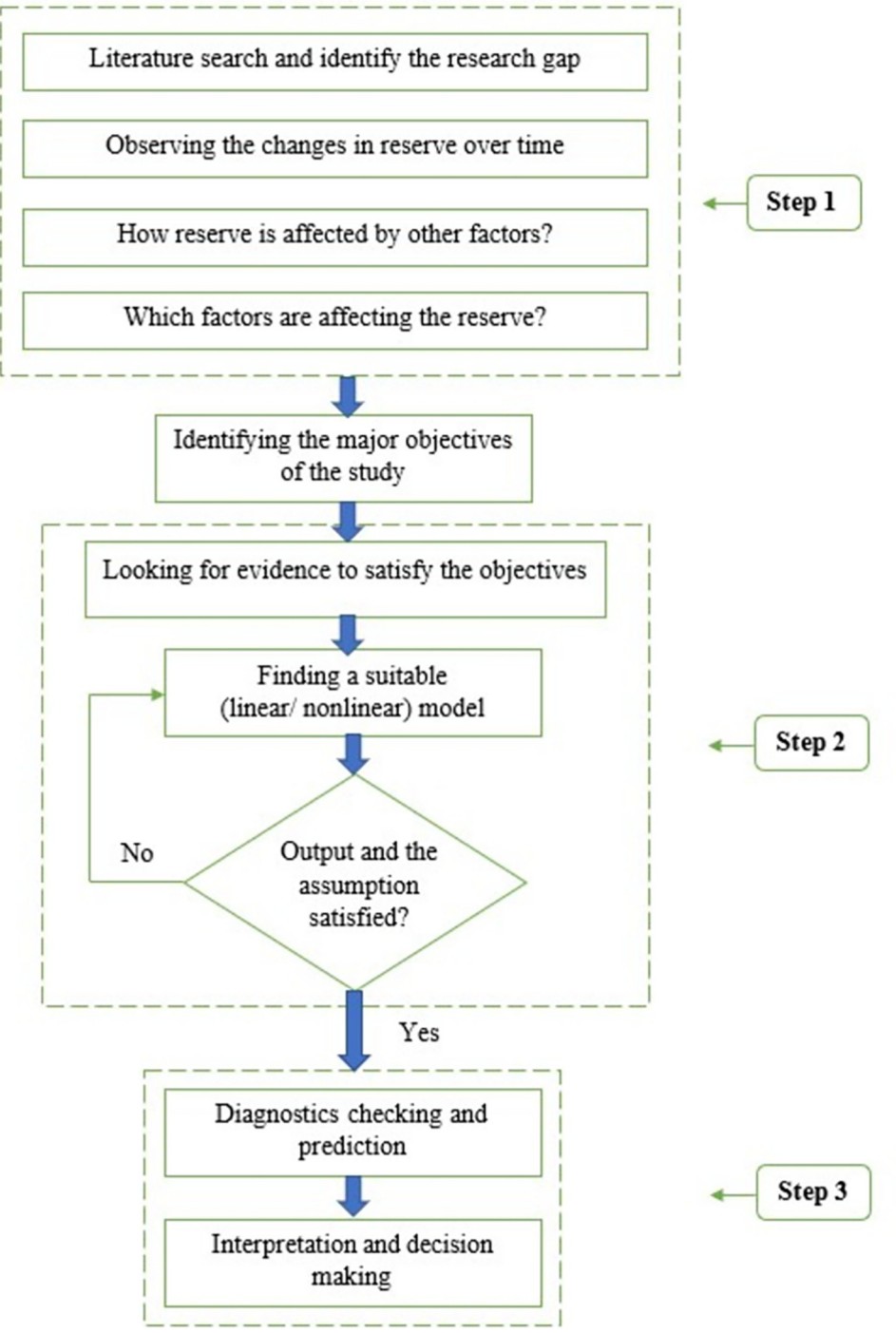

**Fig 1. Research framework.**

For infinitesimal changes in *x*, the effect on *y* is given by the total derivate with respect to *x*: The fact that the change in yield depends on *x* is what makes the relationship between *x* and *y* nonlinear even though the model is linear in the parameters to be estimated.

## Piece-wise polynomial spline

A piecewise polynomial of order of order $k$ with break sequence $\xi$ (necessarily strictly increasing) is any function $f(.)$ that, on each of the half-open intervals $[\varepsilon_j . . \varepsilon_{j+1})$ agrees with some polynomial of degree$< k$. The set of all piecewise polynomial functions of order $k$ with break sequence $\xi$ is denoted here

$$\prod < k, \xi$$

If we simply assign separate cubic polynomials to four intervals of X with cutoffs at $\{C_1, C_2, C_3\}$, we would obtain the fitted values according to

$$\hat{Y} = \begin{cases} \beta_{10} + \beta_{11}X + \beta_{12}X^2 + \beta_{13}X^3 \text{ for } X < c_1 \\ \beta_{20} + \beta_{21}X + \beta_{22}X^2 + \beta_{23}X^3 \text{ for } c_1 \leq X < c_2 \\ \beta_{30} + \beta_{31}X + \beta_{32}X^2 + \beta_{33}X^3 \text{ for } c_2 \leq X \leq c_3 \\ \beta_{40} + \beta_{41}X + \beta_{42}X^2 + \beta_{43}X^2 \text{ for } c_3 \leq X \end{cases}$$

## Generalized additive model (GAM) model

In statistics, a generalized additive model (GAM) is a generalized linear model in which the linear response variable depends on linearly on unknown smooth functions of some predictor variables, and interest focuses on inference about these smooth functions [46]. GAMs were originally developed by Trevor Hastie and Robert Tibshirani to blend properties of generalized linear models with additive models [47]. They can be interpreted as the discriminative generalization of the naive Bayes generative model. In the regression setting, a GAM takes the form:

$$g(E[Y|X_1, X_2, \ldots, X_P]) = \alpha + f_1(X_1) + f_2(X_2) + \cdots + f_p(X_P)$$

here, $Y$ is the continuous response, $X_1, X_2, \ldots, X_P$ are the covariates, and $f_1(.), \ldots, f_p(.)$ are the unspecified smooth (nonparametric) functions.

The model relates a univariate response variable, $Y$, to some predictor variables, $x_i$. An exponential family distribution is specified for $Y$ (for example normal, binomial or Poisson distributions) along with a link function $g$ (for example the identity or log functions) relating the expected value of $Y$ to the predictor variables via a structure such as

$$g(E(Y)) = \beta_0 + f_1(x_1) + f(x_2) + \ldots$$

The functions $f_i$ may be function with a specified parametric form (for example a polynomial, or an un-penalized regression spline of a variable) or may be specified non-parametrically, or semi-parametrically, simply as 'smooth functions', to be estimated by nonparametric means. So, a typical GAM might use a scatterplot smoothing function, such a locally weighted mean, for $f_1(x_1)$, and then use a factor model for $f_2(x_2)$. This flexibility to allow nonparametric fits with relaxed assumptions on the actual relationship between response and predictor provides the potential for better fits to data than purely parametric models, but arguably with some loss of interpretability.

## Results

The authors attempted to explore the trend of the total reserve of Bangladesh and tried to determine the influence of financial and economic indicators on it. Therefore, the authors analyzed the data by exploratory analysis and time series plots to get ideas about the nature of the variables initially. A comparative study is carried out to find the most appropriate model for prediction. The authors believed that the findings of this manuscript will assist the government

**Table 1. Descriptive statistics of the variables included in the study.**

| Variable | Min | Max | Average | Standard Deviation |
|---|---|---|---|---|
| Total Reserve (Response Variable) | 160 | 43172 | 7515 | 11470.33 |
| Total Debt | 4.64 | 38.54 | 15.37 | 9.81 |
| Net Foreign Assets ($) (in billion) | -10.82 | 3117.20 | 479.16 | 812.10 |
| Net domestic credit ($) (in million) | 18941 | 17915237 | 3079067 | 4729485 |
| Inflation, GDP deflator (annual %) | -17.63 | 25.62 | 6.69 | 6.24 |
| Exports of Goods and Services (% of GDP) | 3.40 | 20.16 | 11.03 | 5.09 |
| Imports of Goods and Services (% of GDP) | 11.70 | 27.95 | 17.83 | 4.68 |
| Foreign Direct Investment, Net Inflows | -0.05 | 1.73 | 0.44 | 0.52 |
| GNI ($) | 9.61 | 389 | 51.42 | 100.25 |
| Official Exchange Rate (LCU per) | 15.02 | 84.87 | 49.48 | 23.03 |
| Personal remittances received ($) | 0.02 | 21.75 | 5.15 | 6.18 |

in letting the nation's monetary authority concentrate on how to increase the reserve. The descriptive statistics of the selected variables are presented in Table 1.

Over 44 years of the total reserve the standard deviation is 11470.33 which indicates a high fluctuation in the value. Total debt service (measured as a percentage of exports of goods, services, and primary income) ranges from a maximum of 38.54 to a minimum of 4.04, with an average of 15.37 and a standard deviation of 9.81. The largest Net Foreign Assets (in billions of USD) is 3117.20 and the average Net Foreign Assets is 479 which is not really a big value for a country. Besides the Net Domestic Credit has a large figure compared with NFA. And Personal remittances received in dollars have a peak of 21.75 dollars, a low of 0.02 dollars, an average of 5.148 dollars over a period of 44 years, and a standard deviation of 6.181 dollars. FDI has the lowest standard deviation and NDC has the maximum compared to the other variables [Table 1].

Total reserves comprise holdings of monetary gold, special drawing rights, reserves of IMF members held by the IMF, and holdings of foreign exchange under the control of monetary authorities. The gold component of these reserves is valued at year-end (December 31) London prices. Total reserve is expressed in U.S. dollars [45].

From Fig 2, it is observed that as the year passed by the total reserve of Bangladesh gradually increased since 2001, and it reached to peak in the year 2020 which was 43172 billion US dollars. From 2003 to 2011, there was a sharp upward trend and then it follows a flat line for a few years and started raising till the end of the study period.

According to the top left plot of Fig 3, the overall debt of Bangladesh monotonically decreased from 1986 after reaching its top. The debt was minimum between the years 2015 and 2016 but there was a sudden raise in 2019. The top right plot shows that the net foreign assets keep growing over the whole period of time but it started accelerating from 2010 and hit its highest point of 3117.20 in the year 2020. The net domestic credit is the sum of net claims on the central government and claims on other sectors of the domestic economy (IFS line 32). The net domestic credit has more smooth growth than the net foreign assets. But its acceleration started to a greater extent after 2000 and never went down. The GDP implicit deflator is the ratio of GDP in the current local currency to GDP in constant local currency. Moderate inflation is associated with economic growth, while high inflation can signal an overheated economy. The fluctuation in inflation is observed from the bottom right plot until 1997 and after that, it became quite stable till 2020 [Fig 3]. Moreover, findings depicted that the exports of goods and services, imports of goods and services, and Net inflows have a lot of ups and downs over the period of time. Both these three economic indicators keep increasing till the

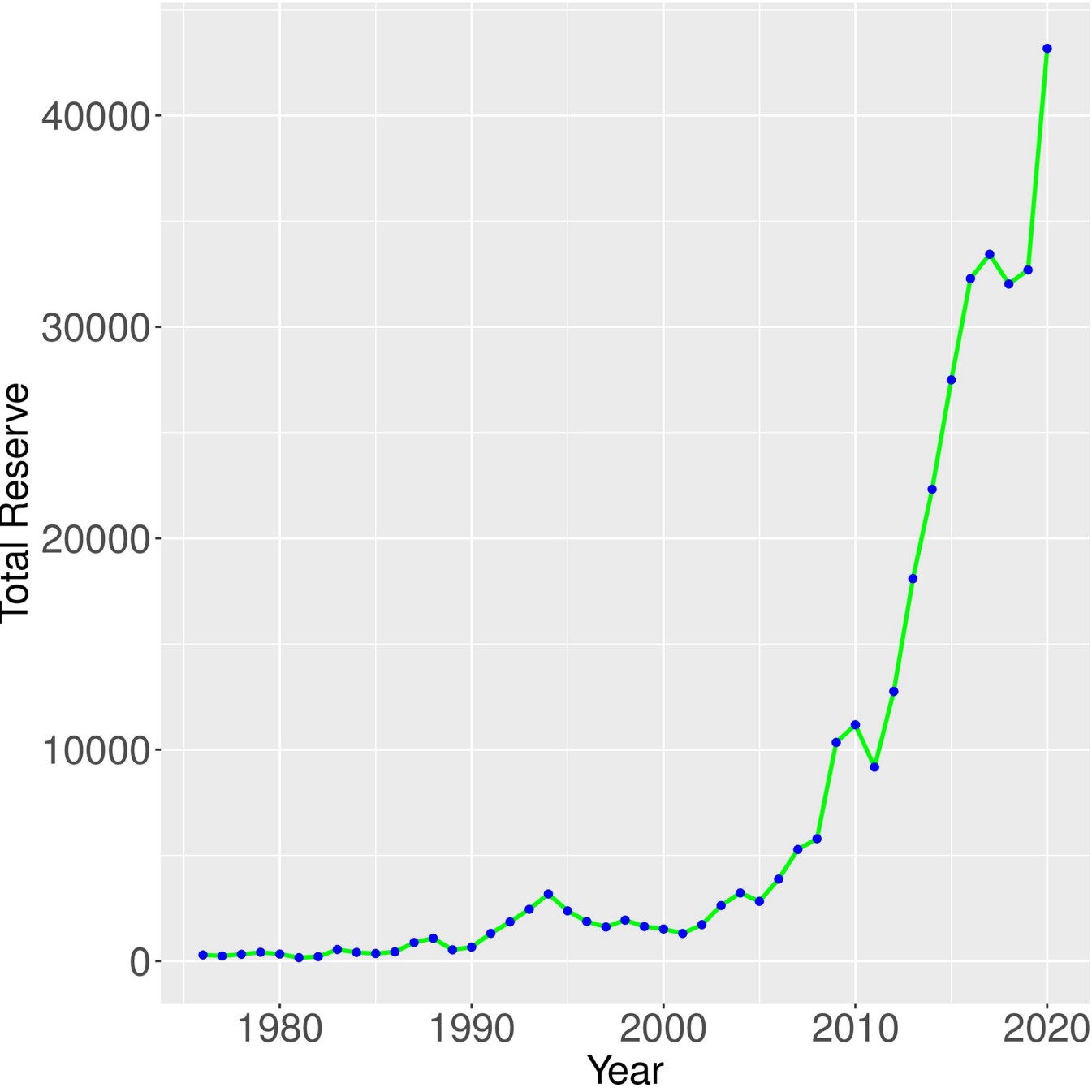

**Fig 2. Time series plot of the response variable (total reserve).**

year between 2012 to 2015 and after reaching the height they started sharply decreasing. On the other hand, Gross national income (GNI), Exchange Rate, and Remittance smoothly keep growing over the whole period of time without any sharp ups and downs. All these three reached their apex in 2020 and counting [Fig 3].

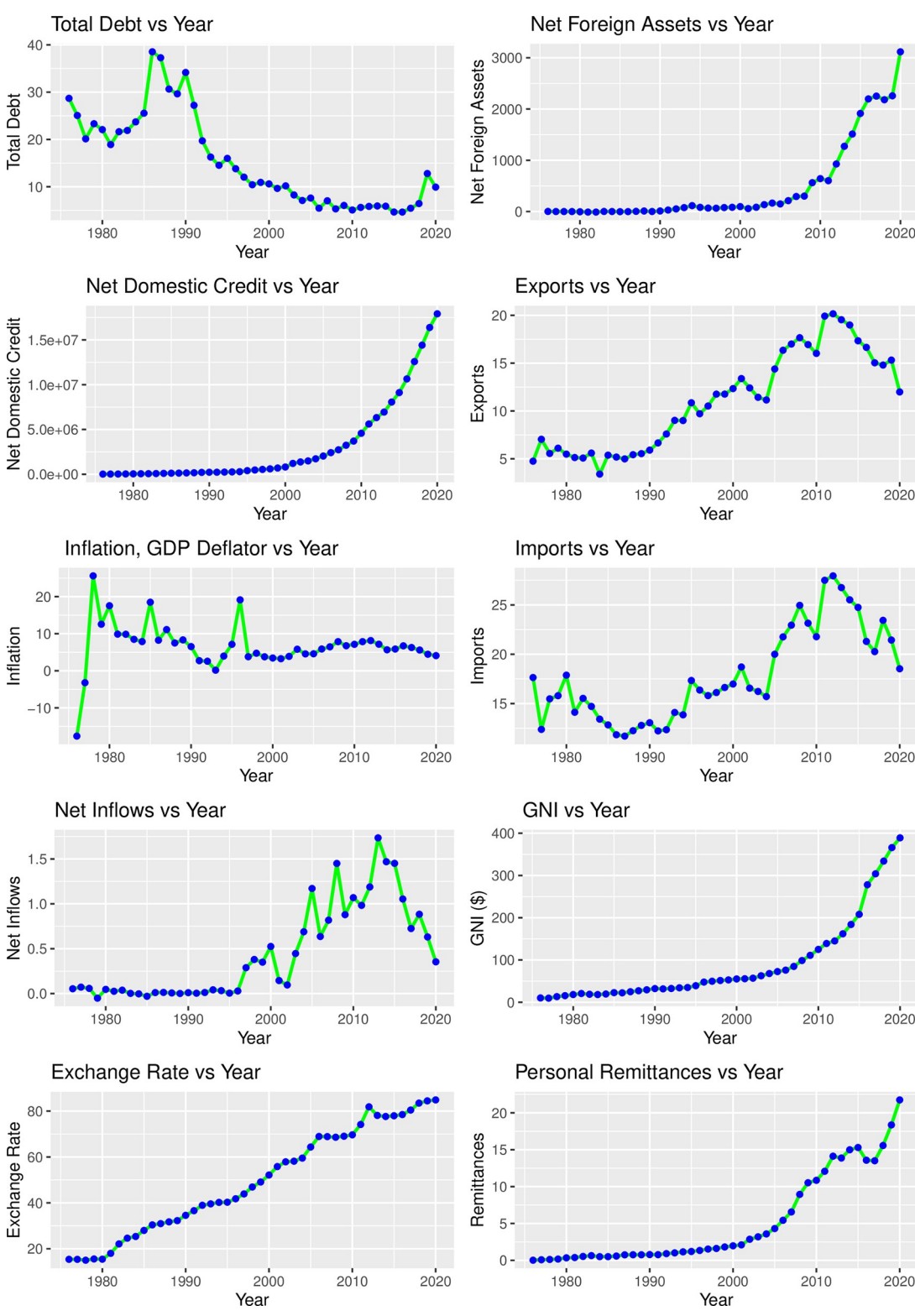

**Fig 3. Time series plot of the selected covariates.**

**Table 2. Estimates of the multiple linear model.**

| Variable | Estimate | Std. Error | t-value | p-value | VIF |
|---|---|---|---|---|---|
| Intercept | -63530 | 183900 | -0.35 | 0.73 | - |
| Year | 32.38 | 93.19 | 0.35 | 0.73 | 256.35 |
| Total Debt | 3.53 | 18.64 | 0.19 | 0.85 | 5.72 |
| NFA | 13.18 | 0.62 | 21.33 | 0.01*** | 43.06 |
| NDC | -0.00099 | 0.0004 | -2.57 | 0.01* | 567.78 |
| Inflation | -8.28 | 13.32 | -0.62 | 0.54 | 1.18 |
| Export | 14.65 | 89.85 | 0.16 | 0.87 | 35.74 |
| Import | -23.64 | 65.65 | -0.36 | 0.72 | 16.16 |
| Net inflows | 169.21 | 409.9 | 0.41 | 0.68 | 7.65 |
| GNI | 48.07 | 17.04 | 2.82 | 0.01** | 499.63 |
| Exchange | -39.07 | 49.56 | -0.79 | 0.44 | 222.98 |
| Remittance | 169.6 | 102.9 | 1.65 | 0.11 | 69.18 |

**Note:** p-value<0.001, '***', 0.001<p< 0.01 '**', 0.01<p<0.05, '*'

The linear model is considered the base model for modeling the observed data. This is because models with linear interdependence on their unknown parameters are easier to fit and simpler to determine the statistical properties of the resulting estimators than models with non-linear dependency on their parameters. Table 2 shows the estimated values of parameters, standard errors, corresponding t-value, p-value, and VIF of the linear model on the response variable total reserve using world bank data.

Then $R^2$ value is 0.9985 and the adjusted $R^2$ value is 0.998, on the other hand from the p-value we can see that only three variable is found significant. This is quite unusual and because of that, we suspected that multicollinearity can be present in the data. To justify our doubt we have calculated the Variance Inflation Factor (VIF) of the variables and added it to the next column of the p-value. From that VIF column, we can see that there is clear evidence of multi-collinearity in the data. It is quite natural for the economic data set that the variables will be correlated as each variable has an effect on the other. The remedial measures for multicollinearity can be applied to the data in many ways but it may reduce the originality of the data and the dependency of each variable is natural and it is important to have it in the data. So before applying any transformation of the variables we have decided to identify the relationship of the covariates with the response variable graphically. We have plotted the response variable total reserve against each of our explanatory variables along with the linear fit and the smooth-fitted line with their 95% confidence interval [S1 Fig].

From S1 Fig, it is clearly visible that the Net Foreign Asset and Net Domestic credit are individually linearly associated with the response but the total debt, inflation, import, and export are showing a non-linear relationship with the total reserve for the data. Moreover, it is observed that the association between the rest of the four covariates and the response is in the same manner. The relationship between the total reserve and foreign direct investment, net inflows (% of GDP) is nonlinear. However, total reserve and gross national income (GNI) continue to be correlated linearly, meaning that as the GNI value climbed, so did total reserve. The official exchange rate of Bangladesh also has nonlinear relation with the total reserve. As seen by curves in the plot, personal remittances received also have a nonlinear relationship with Bangladesh's overall reserve [S1 Fig]. Therefore, it is clearly seen that the suitable model for the data will be a model which incorporates the covariates according to their type of association, which means the combination of linear and non-linear association in the predictor part. For

this reason, we fitted the Generalized Additive Model (GAM) for our data and the findings are presented in Table 3.

The overall GAM model is producing the R-squared value as 0.9985 and we found nine covariates significant where all the linear covariates are significant. In the non-linear part, only the total debt and FDI were found insignificant and the rest of the variables are significant. So, this model solves the problem of multicollinearity found in the linear model. The estimated value of these covariates is given in the first column of the above table. The effective degrees of freedom (edf) is a summary statistic of GAM and it reflects the degree of nonlinearity of a curve [46]. If the value of the edf is equal to 1 then it is equivalent to a linear relationship, if it is between 1 and 2 then it is considered a weakly non-linear relationship, and if edf is greater than 2 then it implies a highly non-linear relationship with respect to the variable. The edf and reference degrees of freedom are one of the main indicators of the goodness of fit of the table. The reflection of these are degrees of freedom also visualized in the bottom figure. From the table and the figure, we can see that most of the edf are very close to the reference df and some of them are equal which is a very good indicator that our model fit is good [Table 3].

With two key exceptions, checking a fitted gam is similar to checking a fitted lm. First, it is important to check the basis dimensions used for smooth terms to make sure they are not arbitrary defaults that force over-smoothing. Secondly, fitting may not always be as robust to violation of the distributional assumptions as would be the case for a regular GLM, so slightly more

**Table 3. Estimates of GAM model.**

| Parametric coefficients | | | | |
|---|---|---|---|---|
| | **Estimate** | **Std. Error** | **t value** | **p-value** |
| **(Intercept)** | 2471.72 | 939.87 | 2.63 | 0.01* |
| **NFA** | 14.43 | 0.35 | 41.49 | <0.001*** |
| **GNI** | -20.78 | 11.09 | -1.87 | 0.07 * |
| **NDC** | 0.0102 | 2.9e-04 | 3.500 | 0.002** |

| Approximate significance of smooth terms | | | | | |
|---|---|---|---|---|---|
| | **Estimates** | **Effective degrees of freedom** | **Reference degrees of freedom** | **F** | **p-value** |
| **s(Year)** | 3066.85 | 1.94 | 1.99 | 19.65 | <0.001*** |
| | -4678.29 | | | | |
| **s(Debt)** | -56.62 | 1.00 | 1.00 | 0.65 | 0.43 |
| | -258.24 | | | | |
| **s(Inflation)** | -1288.39 | 1.91 | 1.99 | 10.04 | <0.001*** |
| | 78.36 | | | | |
| **s(Export)** | 554.62 | 1.13 | 1.47 | 4.45 | 0.04** |
| | 1165.78 | | | | |
| **s(FDI)** | 110.30 | 1.00 | 1.00 | 0.39 | 0.54 |
| | -195.10 | | | | |
| **s(Import)** | 100.25 | 2.83 | 3.32 | 5.44 | <0.001** |
| | -198.05 | | | | |
| | -389.87 | | | | |
| | -1518.10 | | | | |
| **s(Exchange)** | -2790.06 | 1.69 | 1.72 | 24.42 | <0.001*** |
| | -6523.68 | | | | |
| **s(Remittance)** | -3200.27 | 1.98 | 2.00 | 24.43 | <0.001*** |
| | -290.27 | | | | |

**Note:** p-value<0.001, '***', 0.001<p< 0.01 '**', 0.01<p<0.05, '*'

care may be needed here. For instance, since the smoothness selection criterion attempts to lower the scale parameter to the one supplied, un-modeled overdispersion will often result in overfitting. The sensitivity of REML and ML smoothness selection to variations from the presumptive response distribution is also unknown. For these reasons, an enhanced residual quantile-quantile (QQ) plot is used in this technique. According to Wood (2017), one way to determine whether the basis dimension for a smooth is suitable is to estimate the residual variance by comparing residuals that are close neighbors in terms of the smooth's (numeric) variables [46].

Each of these plots offers a unique perspective on the model residuals. Fig 4 depicts the outcomes of the original, inadequately fitted model. A Q-Q plot, located in the top-left corner, contrasts the model residuals with a normal distribution. The residuals of a well-fit model will be almost linear. In contrast, we have a linear line in the plot. There is a residual histogram on the bottom left. This should have a bell-like, symmetrical shape. Now, for our model, we have an approximately bell-shaped histogram. A plot of the residual values is located at the top-right. Around 0, these should be distributed equally but our model values are dispersed around 0 lines. The plot of response against fitted values may be seen on the bottom right. A straight line is formed by a flawless model. Although we don't anticipate a flawless model, we do anticipate that the pattern will cluster around the 1-to-1 line. See that the Q-Q plot no longer curves, the histogram is almost bell-shaped, and the comparison of response vs. fitted values clusters around a 1-to-1 line. These all indicate a much better model fit [Fig 4].

Finally, we compare the linear model and nonlinear model (GAM) with the help of AIC and BIC. Findings revealed that the linear model has AIC 700.33 whereas the nonlinear model (GAM) has AIC 637.78 which is lower. On the other hand, for our linear model, we have a BIC of 723.81 whereas for the GAM model the value of BIC is 670.83. Therefore, both AIC and BIC value for GAM is lower which indicate a better fit for our corresponding data. Fig 5 compares the observed and fitted values for Bangladesh's total reserve determined using the GAM model, along with the 95% confidence interval. It has been noted that there is very little variation between the observed value and the predicted value, indicating that the model fits the data quite well.

## Discussion

The study mainly focused on finding the effecting determinants of reserve based on the World Bank data but without a suitable model, it is not possible. Our chosen model (GAM) provides a satisfactory fit for the data based on the model choice criteria. Referring to a study done in Indonesia by [48], where they included the variables of foreign debt, exchange rate, inflation, and exports as explanatory factors to express the reserve using time-series data acquired from the International Monetary Funds (IMF), the Central Bureau of Statistics (BPS), and the Central Bank of Indonesia between January 2016 and December 2018 and implemented the Auto-regressive Distributed Lag technique. According to their findings, the simultaneous volatility of Indonesia's foreign exchange reserves is highly influenced by exports, inflation, exchange rates, and foreign debt and their finding quite match ours. Researchers also started working on foreign reserve based in Bangladesh and found that exchange rate, home interest rate, remittance, import, and export plays a vital role in expressing foreign reserve but not foreign aid [13]. A previous study pointed out that commodity exports may influence reserve accumulation [49]. According to our non-linear model, the total debt and FDI did not found to be influential factors for reserve. A study pointed out that trade openness and the wide money-to-GDP ratio are the two main factors that have a positive correlation with the number of reserves. The demand for reserves, however, appears to be reduced by financial progress [50].

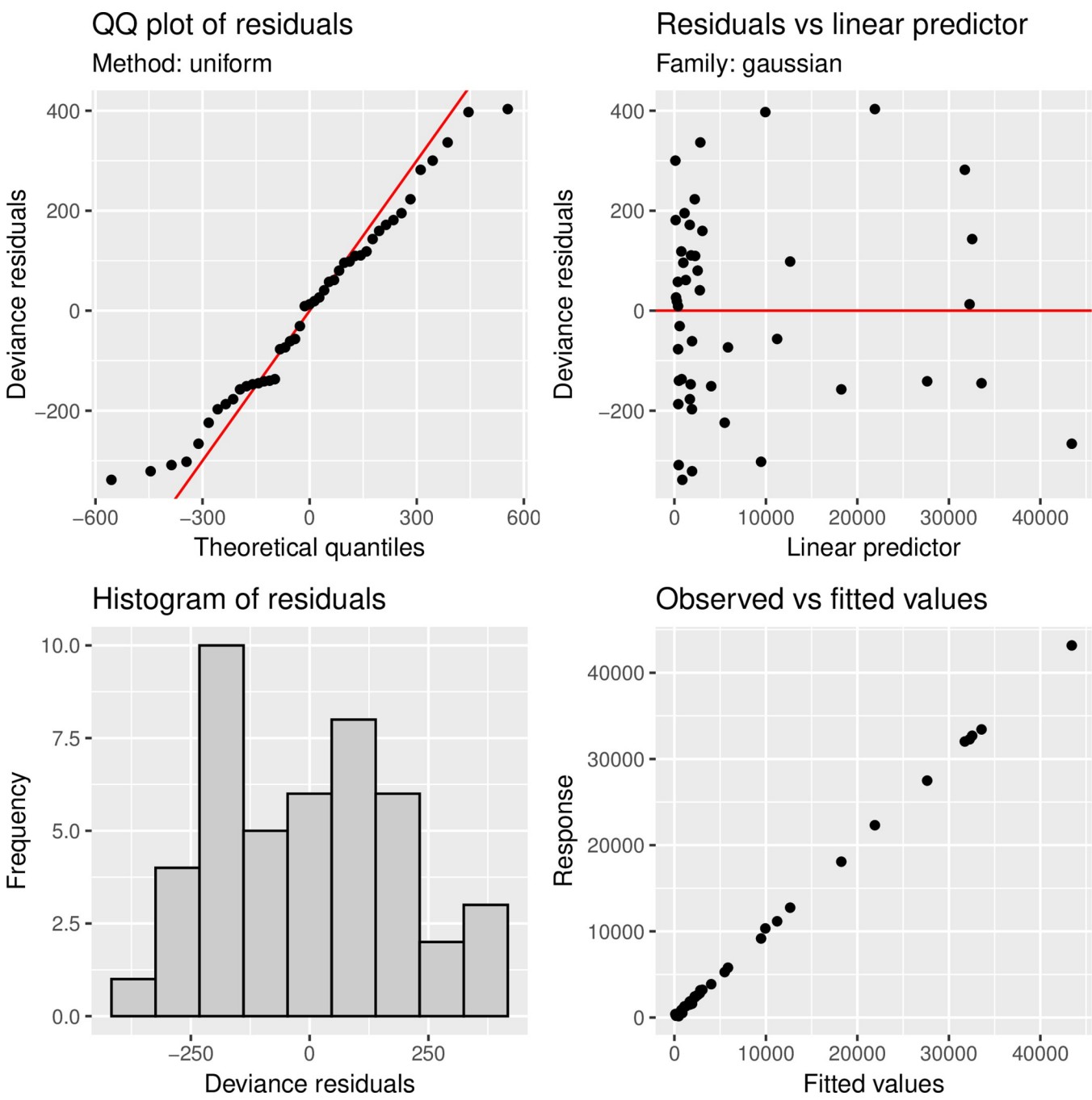

**Fig 4. Diagnostic plots of the fitted GAM model.**

The annual import, export, inflation, remittance, and exchange rate are the factors that are non-linearly affecting the total reserve, and net foreign asset, net domestic credit, and GNI is affecting the total reserve in a linear manner according to our study. It is observed that most of the research is based on the only foreign reserve where we have tried to focus on total reserve and time series models are preferred as the analysis tools. As we found evidence of non-linearity in the data, so we preferred to choose a model in a manner that can find the appropriate determinants as well as express the type of relationship the covariates hold with the response.

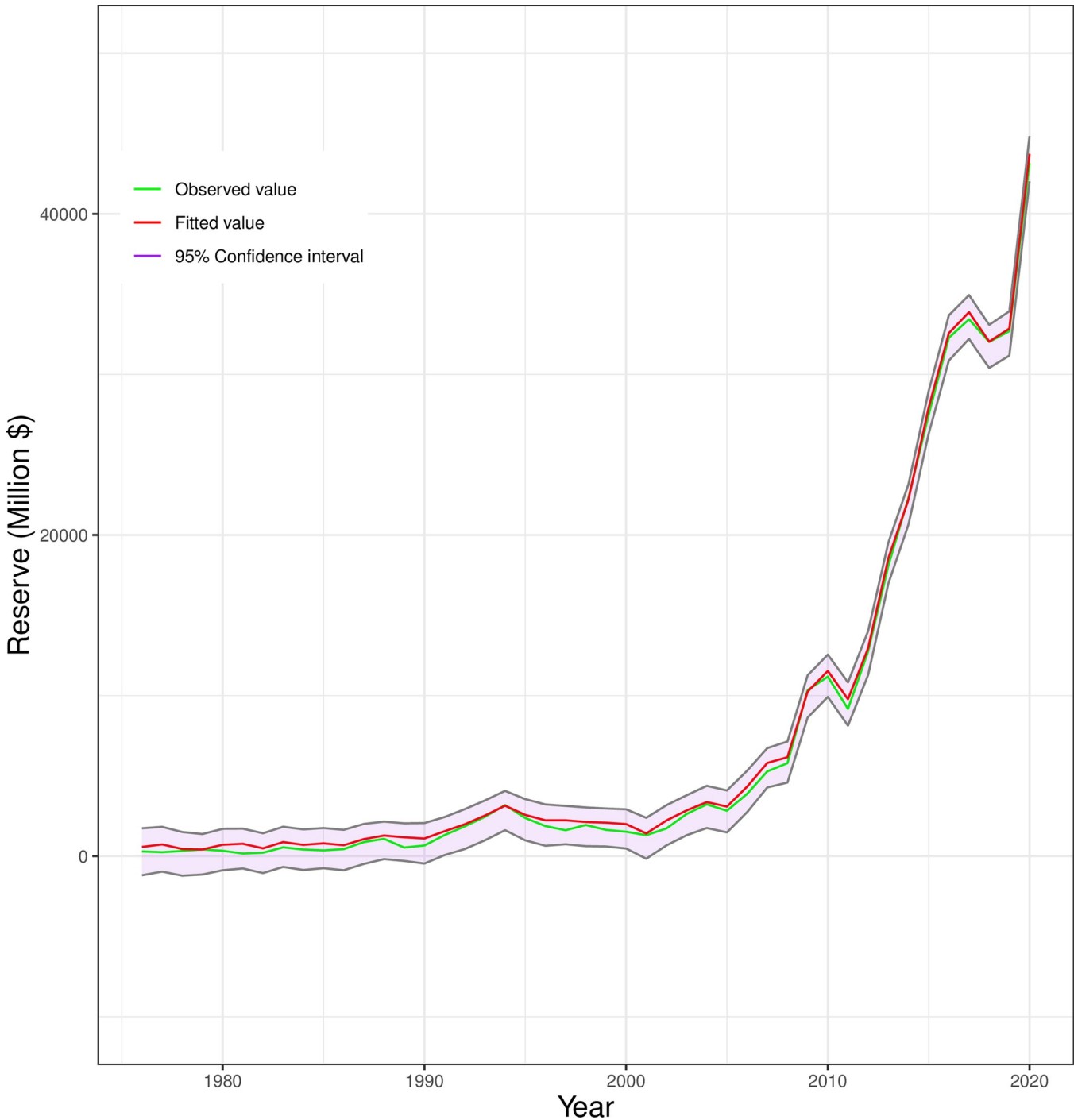

**Fig 5. Comparison of predicted and observed total reserve in Bangladesh.**

## Policy suggestions

Financial development, also referred to as financial globalization, plays a crucial role in the rise in investment efficiency, commercial opportunities, commerce in products and services, and technical advancement [51]. This sign required exercising more caution in order to improve

this financial situation and prepare for any impending financial emergency. Because of this, the government and decision-makers can concentrate on the elements that we have determined to be important for Bangladesh's reserve. Additionally, while the dollar reserves are being depleted, the government and central bank should implement a number of measures to curb rising import costs, such as limiting the imports of non-essential goods. In addition, for Bangladesh to return to a balanced position, more foreign aid must be provided, more domestic investment must be encouraged to raise GNI, and prompt repayment of loans made to the private sector from the foreign exchange reserve must be made.

## Conclusions

The authors aimed to fit a non-linear model (GAM) from the graphical representation that our response (total reserve) is largely connected to covariates nonlinearly. But for the covariates (NFA, NDC, and GNI), it is observed that a linear relationship with the target variable, which we included as linear in the model. From the fitted model, we found a significant relationship between the response and covariates. In the case of multiple regression, net foreign assets, net domestic credit, exports of goods and services (% of GDP), imports of goods and services (% of GDP), inflation, gross national income (current US dollars), an official exchange rate (LCU per US dollars, period average), and personal remittances received (current US dollars) and total debt, net inflows are insignificant, however, Total Debt and net Inflows are found insignificant for our model. Moreover, the linear model was suffering from multicellularity. Given that the GAM model's AIC and BIC values are lower than those of the linear model, this suggests a better fit for the model i.e., the performance of the nonlinear model is superior to the linear model. On the other hand, the value of the adjusted $R$-square of GAM is likewise larger than that of the linear model, indicating that GAM provides a more comprehensive explanation of the data. Therefore, the authors recommended to use the GAM model instead of the linear model for modelling the total reserve of Bangladesh.

## Supporting information

**S1 Fig. Total reserves vs the selected covariates along with the fitted lines and 95% confidence intervals.**
(TIF)

## Author Contributions

**Conceptualization:** Md. Sifat Ar Salan, Md. Moyazzem Hossain, Mohammad Alamgir Kabir, Ajit Kumar Majumder.

**Data curation:** Md. Sifat Ar Salan, Mahabuba Naznin, Bristy Pandit, Imran Hossain Sumon.

**Formal analysis:** Md. Sifat Ar Salan, Mahabuba Naznin, Md. Moyazzem Hossain.

**Methodology:** Md. Sifat Ar Salan, Md. Moyazzem Hossain.

**Supervision:** Md. Moyazzem Hossain, Mohammad Alamgir Kabir, Ajit Kumar Majumder.

**Visualization:** Md. Sifat Ar Salan, Mahabuba Naznin, Bristy Pandit.

**Writing – original draft:** Md. Sifat Ar Salan, Mahabuba Naznin, Bristy Pandit, Imran Hossain Sumon, Md. Moyazzem Hossain.

**Writing – review & editing:** Md. Sifat Ar Salan, Md. Moyazzem Hossain, Mohammad Alamgir Kabir, Ajit Kumar Majumder.

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
