## [Decision Letter · Decision Letter 0]

3 Jan 2023

PONE-D-22-28792Relationships Between Total Reserve and Financial Indicators of Bangladesh: Application of Generalized Additive ModelPLOS ONE

Dear Dr. Hossain,

Thank you for submitting your manuscript to PLOS ONE. After careful consideration, we feel that it has merit but does not fully meet PLOS ONE’s publication criteria as it currently stands. Therefore, we invite you to submit a revised version of the manuscript that addresses the points raised during the review process.

We look forward to receiving your revised manuscript.

Kind regards,

Roni Bhowmik, Ph.D.

Academic Editor

PLOS ONE

Journal Requirements:

3. We note you have included a table to which you do not refer in the text of your manuscript. Please ensure that you refer to Table 2 in your text; if accepted, production will need this reference to link the reader to the Table.

Additional Editor Comments:

Dear Author(s),

Please consider any outstanding revision requests from all reviewers, including the reviewers who recommended rejection.

You can respond to the comments in this thread and resubmit the revised manuscript. We encourage you to submit your revised manuscript with tracked changes to facilitate the review.

Thank you for your time and consideration,

Regards-

Dr. Roni Bhowmik

Associate Editor

Reviewers' comments:

Reviewer's Responses to Questions

**Comments to the Author**

1. Is the manuscript technically sound, and do the data support the conclusions?

Reviewer #1: No

Reviewer #2: Partly

Reviewer #3: No

2. Has the statistical analysis been performed appropriately and rigorously? 

Reviewer #1: I Don't Know

Reviewer #2: Yes

Reviewer #3: No

3. Have the authors made all data underlying the findings in their manuscript fully available?

Reviewer #1: Yes

Reviewer #2: Yes

Reviewer #3: No

4. Is the manuscript presented in an intelligible fashion and written in standard English?

Reviewer #1: Yes

Reviewer #2: Yes

Reviewer #3: No

5. Review Comments to the Author

Reviewer #1: It is unclear what is the motivation of this study. Indeed, this study has identified some anomalies (net outcome), which I have identified in my summary above. One can find so many things through big data analytics, which are sometimes interesting. However, in academic study, there are two broad approaches. First, the study could identify a research question based on a debate in the literature, identify the knowledge gap, and draw a hypothesis to test. Second, based on conjecture or general observation, do empirical tests, and find results that can be explained deeply in the context of theory and broader literature. I see neither approach in this study.

Reviewer #2: Dear authors,

I'd like to congratulate you and your team on your excellent research work in your paper submitted for publication in this prestigious journal. The topic is very interesting, and I enjoyed it. I would like to thank you for your efforts in presenting your research work in such a professional manner. However, before your work is recommended or accepted, a few comments must be included/ addressed to improve the quality of your work as well as for future publication in this reputable journal. I have the following observations, questions, and comments that may help to improve your work. The authors must modify the following points in great detail.

1. In the abstract, please include 2-3 special quantitative achievements from the findings of this study in the context of the environment by combining the research objectives and problems. Please limit your abstract to 250 words. Check spellings for many words that are misspelt or written in haste.

2. The introduction section needs a few more sentences to strengthen the article, and please include the research problem, objective, and novelty in the last paragraph of the Introduction section.

3. Include a few more sentences at the beginning of the introduction explaining your paper's contribution to the environment, climate change impact, and sustainability, as well as your attempts to deal with or present solutions to a specific problem/s and your unique contribution with this research paper.

4. Please also present the methodology section in a concise graphical format.

5. The literature review section is very weak; please revise it.

6. Please present your literature review in the form of a SmartArt chart.

7. Just after the Methodology, please mention the societal benefits of your research in terms of evaluating its key determinant.

8. In 500-750 words, explain research problems, solutions, and the theoretical contribution of your study in the "Results" section.

9. Please include graphical presentations of your findings.

10. Describe why you placed this study in a separate section of "Policy Suggestions" just before the section of "Conclusions."

I found that the literature section is a little weak, shift your study a little more towards environment friendly and sustainability, therefore it requires more studies to be reviewed therefore I suggest you to include the following work:

https://doi.org/10.1016/j.jclepro.2021.128585

https://doi.org/10.1016/j.jclepro.2021.128109

https://doi.org/10.1016/j.heliyon.2021.e06952

https://doi.org/10.1007/s11356-022-20567-6

https://doi.org/10.1177/09763996211041215

https://doi.org/10.1007/s11356-021-15421-0

https://doi.org/10.1007/s11356-021-14745-1

https://doi.org/10.1007/s11356-021-13441-4

https://doi.org/10.1007/s10668-021-01418-9

https://doi.org/10.1016/j.techfore.2022.121524

https://doi.org/10.1016/j.resourpol.2022.102612

I think above all studies will make this study more relevant in bridging the gap with literature.

Looking forward for your revised submission.

Reviewer #3: The topic is interesting. However, the paper looks like a technical report more than a research article. The manuscript has not been written up to the standards of the journal. I am not sure about the goal of the paper. It seems that mathematical narration is fine. The paper is a statistical practice without a well-justified motivation. Motivation is weak. The selection of variables is arbitrary rather than theoretical and empirical consideration. The paper cannot persuade the readers due to weak evidences and arguments in designing research model. Overall, I am afraid, I cannot recommend it for publication in PloS ONE.

6. PLOS authors have the option to publish the peer review history of their article (what does this mean?). If published, this will include your full peer review and any attached files.

Reviewer #1: No

Reviewer #2: **Yes: **Vishal

Reviewer #3: No

---

## [Author Response · Author response to Decision Letter 0]

22 Jan 2023

Dear Editor,

We would like to express our sincere gratitude to the three reviewers and the Academic Editor for their valuable comments. We have considered all the comments made by the reviewers and thoroughly revised and formatted the manuscript accordingly. A detailed response to each of the comments is provided below.

Author's response to the Academic Editor comments:

1. Thank you very much. The required files are submitted through the submission system. 

We include all required information in the cover letter. Revised texts are in red color. 

Author's response to the Journal Requirements:

1. Many thanks. We revised the manuscript following the PLOS ONE style. 

2. Thanks. We update the data availability statement. The data is freely available on World Development Indicators of World Bank database. Data can be access through the link: https://data.worldbank.org/country/BD

Revised texts are in red color. Page: 20

3. Thanks. We refer Table 2 in the texts. Revised texts are in red color. 

Page: 12

4. Thank you very much. We add the caption of Supporting Information files at the end of the manuscript and update the citation in texts. Revised texts are in red color. Page: 19

Author's response to the Reviewer 1 comments:

Thank you very much for your comments. We revise the manuscript incorporating the research gap, revised the objectives, and updated the methodology and result sections. Revised texts are in red color. 

Page: 1-19

Author's response to the Reviewer 2 comments:

1. Thank you very much for your comments and feedback. We believe that it helps to improve the quality of the manuscript. 

As we know, POLS ONE accepts up to 500 words in abstract. In our case, it consists of 373 words. We also check the spelling. Revised texts are in red color. Page: 1-2

2. Thanks. We revised the Introduction section. Revised texts are in red color. 

Page: 3-5

3. Thank you very much. Actually, this manuscript does not consider the climate variables, however, we revise the manuscript and tried to link economic growth and economic development. Revised texts are in red color. 

Page: 4-5

4. Thank. We add a figure to present the research framework. Revised texts are in red color. 

Page: 5-6

5. Thanks. We revised the manuscript. Revised texts are in red color. 

Page: 2-5

6. Thank you very much. We add a flowchart in the Methodology section. Revised texts are in red color. 

Page: 5-6

7. Thanks. We add it in the Methodology section. Revised texts are in red color. 

Page: 5

8. Thanks. Actually, it is an applied work. The authors tried to find the most appropriate model to predict the total reserve. The result section is revised as per your suggestion. Revised texts are in red color. 

Page: 9, 17

9. Thanks. We add a figure to present the final findings of this manuscript. Revised texts are in red color. 

Page: 17

10. Thanks. We revised the manuscript in light of the suggested papers and cite them. Revised texts are in red color. 

Page: 3-5

Author's response to the Reviewer 3 comments:

Thank you very much for your comment and feedback. We revise the motivation and objectives of this manuscript. We hope the revised version will satisfy your expectation. Revised texts are in red color. 

Page: 1-19

Finally, the revised manuscript has been produced following the valuable comments and suggestions of the reviewers. Once again, we would like to thank the reviewers for their sincere dedication, professional insights, and earnest cooperation in reviewing the manuscript.

---

## [Decision Letter · Decision Letter 1]

27 Feb 2023

PONE-D-22-28792R1Relationships Between Total Reserve and Financial Indicators of Bangladesh: Application of Generalized Additive ModelPLOS ONE

Dear Dr. Hossain,

Thank you for submitting your manuscript to PLOS ONE. After careful consideration, we feel that it has merit but does not fully meet PLOS ONE’s publication criteria as it currently stands. Therefore, we invite you to submit a revised version of the manuscript that addresses the points raised during the review process.

I find your work has a very minimal contribution to the introduction, literature, and methodology part. Please consider all outstanding revision requests from both reviewers, including the reviewers who recommended rejection. As, I don’t think your contribution is good enough to be published in a very high standard journal like the Plos One. You can respond to the comments in this thread and resubmit the revised manuscript. We encourage you to submit your revised manuscript with tracked changes to facilitate the review.

We look forward to receiving your revised manuscript.

Kind regards,

Roni Bhowmik, Ph.D.

Academic Editor

PLOS ONE

Additional Editor Comments :

Dear Authors,

I find your work has a very minimal contribution to the introduction, literature, and methodology part. Please consider all outstanding revision requests from all reviewers, including the reviewers who recommended rejection. As, I don’t think your contribution is good enough to be published in a very high standard journal like the Plos One.

You can respond to the comments in this thread and resubmit the revised manuscript. We encourage you to submit your revised manuscript with tracked changes to facilitate the review.

Reviewers' comments:

Reviewer's Responses to Questions

**Comments to the Author**

1. If the authors have adequately addressed your comments raised in a previous round of review and you feel that this manuscript is now acceptable for publication, you may indicate that here to bypass the “Comments to the Author” section, enter your conflict of interest statement in the “Confidential to Editor” section, and submit your "Accept" recommendation.

Reviewer #2: All comments have been addressed

Reviewer #3: All comments have been addressed

2. Is the manuscript technically sound, and do the data support the conclusions?

Reviewer #2: Yes

Reviewer #3: No

3. Has the statistical analysis been performed appropriately and rigorously? 

Reviewer #2: Yes

Reviewer #3: No

4. Have the authors made all data underlying the findings in their manuscript fully available?

Reviewer #2: Yes

Reviewer #3: No

5. Is the manuscript presented in an intelligible fashion and written in standard English?

Reviewer #2: Yes

Reviewer #3: Yes

6. Review Comments to the Author

Reviewer #2: Dear Authors,

I would like to congratulate you and your team for doing such a good research work in your submitted paper. Topic is very interesting and I liked the topic and appreciate your efforts to present your revised research work in such a nice manner. I am satisfied from your efforts you employed in the revision and I found all my suggested comments have been incorporated or addressed perfectly. Therefore, I strongly recommended this article for acceptance for further publication in this reputed journal without any more changes.

The revised version of the paper looks perfect, in the section of the literature review the very relevant, connected, and updated with new references.

Methodology mentions this research's socio-economic benefits in evaluating its crucial determinant.

The section on results in the revised version explains all of the tables more briefly, and the explanations for each table in the "Results" sections, define more visibility of empirical results.

In the revised version after adding about 150 words to the conclusion section related to policy implications explaining and connecting the future scope of your research study, any limitations encountered while conducting your research, and the procedure for removing research limitations.

The revised version communicates the introduction which provides helpful information about the intended topic; however, it needed to be revised to make it meaningful and the authors are successfully able to revise it. Indeed, the research GAP is now well explained in a scientific way. Therefore, the research can be recommended for publication without any more modification in its the current form which provides sufficient justification and arguments.

Reviewer #3: The revised version of the manuscript seems to contain the previous mistakes and shortcomings event though the author attempt to the response my queries. In general, the paper looks like a report paper or a project to test out existing empirical methods on different quantitative techniques rather than an academic paper. This paper does not meet the requirements for a scientific article. The motivation is still not strong enough. Literature review section is missing. Theoretical background of selection of the variables is missing. Therefore sorry for inconvenience and harsh decision from my side. I do NOT recommend this work for publication in as a prestigious journal as PloS ONE.

7. PLOS authors have the option to publish the peer review history of their article (what does this mean?). If published, this will include your full peer review and any attached files.

Reviewer #2: **Yes: **Vishal

Reviewer #3: No

---

## [Author Response · Author response to Decision Letter 1]

10 Mar 2023

Dear Editor,

We would like to express our sincere gratitude to the three reviewers and the Academic Editor for their valuable comments. We have considered all the comments made by the reviewers and thoroughly revised and formatted the manuscript accordingly. A detailed response to each of the comments is provided below:

Response to the Academic Editor comments: 

Thank you very much. The required files are submitted through the submission system. 

We include all required information in the cover letter. Revised texts are in red color. 

Many thanks. We revised the manuscript as per the review comments. Revised texts are in red color. 

Page: 2-6

Response to the Reviewer 2 comments: 

Thank you very much for your positive as well as detailed comments. 

We are also thankful to you for recommending the manuscript for publication without any more modification in its current form. 

Response to the Reviewer 3 comments: 

Thank you very much for your comments and feedback. We believe that your negative comments help to improve the quality of the manuscript. 

The objective of this paper is to identify the nature of the relationship and influence of economic indicators on the total reserve of Bangladesh using a suitable statistical model. 

From our previous experience of publishing papers in PloS ONE as well as the guidelines of this journal, we format this paper. 

We revise the motivation of this study.

Basically, we merged the Background and Literature review following many of the published papers in PloS ONE. As per your comments, we add the literature review section separately. 

The selection of carraraites is based on the existing literature review, data availability, and self-efficacy of the authors. 

Revised texts are in red color. Page: 4-6 

Finally, the revised manuscript has been produced following the valuable comments and suggestions of the reviewers. Once again, we would like to thank the reviewers for their sincere dedication, professional insights, and earnest cooperation in reviewing the manuscript.

---

## [Decision Letter · Decision Letter 2]

27 Mar 2023

Relationships Between Total Reserve and Financial Indicators of Bangladesh: Application of Generalized Additive Model

PONE-D-22-28792R2

Dear Dr. Hossain,

We’re pleased to inform you that your manuscript has been judged scientifically suitable for publication and will be formally accepted for publication once it meets all outstanding technical requirements.

Kind regards,

Roni Bhowmik, Ph.D.

Academic Editor

PLOS ONE

Additional Editor Comments (optional):

Reviewers' comments:

Reviewer's Responses to Questions

**Comments to the Author**

1. If the authors have adequately addressed your comments raised in a previous round of review and you feel that this manuscript is now acceptable for publication, you may indicate that here to bypass the “Comments to the Author” section, enter your conflict of interest statement in the “Confidential to Editor” section, and submit your "Accept" recommendation.

Reviewer #1: All comments have been addressed

Reviewer #3: All comments have been addressed

2. Is the manuscript technically sound, and do the data support the conclusions?

Reviewer #1: Partly

Reviewer #3: Yes

3. Has the statistical analysis been performed appropriately and rigorously? 

Reviewer #1: I Don't Know

Reviewer #3: Yes

4. Have the authors made all data underlying the findings in their manuscript fully available?

Reviewer #1: No

Reviewer #3: Yes

5. Is the manuscript presented in an intelligible fashion and written in standard English?

Reviewer #1: Yes

Reviewer #3: Yes

6. Review Comments to the Author

Reviewer #1: (No Response)

Reviewer #3: I have completed my review of the study and determined that the paper has been improved. I have decided that the article is suitable for publication.

7. PLOS authors have the option to publish the peer review history of their article (what does this mean?). If published, this will include your full peer review and any attached files.

Reviewer #1: No

Reviewer #3: No

---

## [Editor Report · Acceptance letter]

30 Mar 2023

PONE-D-22-28792R2 

Relationships Between Total Reserve and Financial Indicators of Bangladesh: Application of Generalized Additive Model 

Dear Dr. Hossain:

I'm pleased to inform you that your manuscript has been deemed suitable for publication in PLOS ONE. Congratulations! Your manuscript is now with our production department. 

Kind regards, 

on behalf of

Associate Professor Roni Bhowmik 

Academic Editor

PLOS ONE